# Demand prediction for shared bicycles around metro stations incorporating STAGCN

**Xue Xing**[iD]*, **Le Wan**[iD], **Fahui Luo**

School of Information and Control Engineering, Jilin University of Chemical Technology, Jilin, China

* xingx@jlict.edu.cn

## Abstract

The seamless integration of shared bikes and metro systems promotes green and eco-friendly travel, yet the supply-demand imbalance of shared bikes around metro stations remains a critical challenge, making accurate demand prediction particularly crucial. Targeting metro-adjacent areas, this study proposes a method to identify shared bike trips connecting to metro usage, effectively filtering out approximately 24% of non-connecting travel records within the buffer zones. A predictive model integrating a Spatiotemporal Attention Graph Convolutional Network (STAGCN), Long Short-Term Memory (LSTM) network, and Informer is developed to forecast shared bike demand for metro connectivity. Specifically, the Informer model incorporates STAGCN to capture spatial correlations in bike demand and introduces an LSTM module to learn long- and short-term temporal dependencies. The final demand prediction is generated through a multilayer perceptron. Experiments conducted on shared bike and metro datasets in Shenzhen demonstrate that the proposed model achieves a coefficient of determination ($R^2$) of 0.893, outperforming baseline models by 6.7% in prediction accuracy. Additionally, it exhibits lower Root Mean Square Error (RMSE) and Mean Absolute Error (MAE) compared to traditional time-series forecasting methods. The proposed demand prediction model can assist operators in optimizing the allocation of shared bike resources, which is of great significance for improving user experience.

## 1 Introduction

As the economy grows and urbanization speeds up, traffic jams, pollution, and energy shortages have become more severe. Urban rail transit plays a key role in solving these problems and promoting social stability. In recent years, the scale and coverage of urban rail transit networks in major cities have expanded and become more stable. With its advantages of large capacity, speed, and punctuality, urban rail transit has become a major mode of transportation for citizens. However, it needs to be integrated with other transport modes to form a complete travel chain.

**Data availability statement:** The dataset required for the results of this study has been uploaded to a stable, publicly accessible repository. The website address is https://doi.org/10.34740/kaggle/dsv/12046915.

**Funding:** Science and Technology Development Plan Project of Jilin Province (20210101416JC) Education Industrial Cultivation Project of Jilin Province (JJKH2023030 Y) The funders had no role in study design, data collection and analysis, decision to publish, or preparation of the manuscript.

**Competing interests:** The authors have declared that no competing interests exist.

Shared bikes, a convenient transport mode, have become popular in many Chinese cities [1]. Unlike traditional station-based shared bikes, dockless ones can be parked anywhere, which has made them the mainstream in China [2]. Shared bikes complement urban rail transit, easing the "first-mile" and "last-mile" problem. Exploring the spatio-temporal patterns of integrated bike-sharing-metro mobility systems holds significant value in revealing the complex dynamic mechanisms underlying multimodal transportation integration. This investigation facilitates the optimization of resource allocation in public transit networks, enhances transfer service efficiency through improved network synergies, and ultimately promotes sustainable urban mobility characterized by green travel behaviors. Consequently, this research domain has progressively emerged as a focal point in urban transportation planning and sustainable mobility studies, attracting growing scholarly attention to its theoretical frameworks and practical implementations [3–5]. Yet, the current distribution of shared bikes around metro stations is mismatched with actual use. Some stations have insufficient supply during peak hours, forcing users to seek other transport, while others have excess bikes that go unused, wasting public space and resources. Therefore, accurately predicting shared bike demand around metro stations is crucial for optimizing resource allocation and enhancing user experience. Scientific prediction can ensure enough bikes are available at high-demand stations during peak hours and prevent over-supply at low-demand stations, improving usage efficiency and operational benefits, and promoting sustainable urban transport development.

Currently, demand prediction for shared bikes primarily relies on data analytics and machine learning technologies. Through the analysis and mining of historical usage data, operational patterns and influencing factors can be identified to forecast future bike-sharing demands. Machine learning techniques enable the construction of predictive models that enhance forecasting accuracy and reliability [6]. However, predicting metro-integrated shared bike demand still faces three key challenges: First, due to the difficulty in obtaining bike-sharing and metro-access trip records, simply equating bike-sharing trips within metro station vicinities to metro-access trips constitutes a methodological limitation [7–9]. This approach incorrectly incorporates numerous bike-sharing activities unrelated to integrated metro services into multimodal mobility analyses, resulting in significant discrepancies between research findings and users' actual travel patterns. Therefore, identifying metro-access trip data is a prerequisite for predicting bike-sharing demand oriented toward metro transfers. Secondly, the prediction model needs to account for multiple interdependent factors influencing demand around metro stations, including but not limited to urban spatial configuration, population density, and transportation network characteristics. Thirdly, the spatiotemporal characteristics of demand require sophisticated modeling approaches, as usage patterns exhibit significant variations across different time periods (peak/off-peak), weather conditions, and geographical locations.

Addressing these predictive challenges holds substantial practical significance. Accurate demand forecasting could provide theoretical foundations for optimizing bike redistribution strategies, effectively resolving haphazard parking issues near

metro stations. More importantly, it ensures better alignment between supply and the actual last-mile connectivity needs, thereby elevating urban mobility efficiency. This technological breakthrough will ultimately facilitate the seamless integration of dockless bike-sharing systems into multimodal transportation networks, creating a more sustainable urban commuting ecosystem.

The primary contributions of this study are threefold, addressing key gaps in bike-sharing demand prediction, particularly for metro-connection scenarios:

(1) Addressing the Challenge of Trip Identification: Existing methods struggle to accurately identify genuine bike-metro connecting trips from large-scale spatio-temporal data, often leading to noisy or biased demand datasets. To address this, we propose a novel identification framework. This framework establishes optimal, data-driven buffer zones around metro stations and develops an effective filtering mechanism based on spatio-temporal patterns to isolate true connecting records. This provides a more reliable and robust foundation for studying and predicting metro-connection demand, a critical step often overlooked or crudely handled in prior research.

(2) Advancing Metro-Connection Demand Prediction: While spatial-temporal dependencies are crucial for demand prediction, existing models often inadequately capture the complex interplay between localized temporal patterns around stations and the broader spatial interactions across the metro network for connection-specific demand. To bridge this gap, we introduce a novel bike-sharing demand prediction model specifically designed for metro-connection scenarios. Our model significantly enhances the Informer architecture by integrating a Spatial-Temporal Attention Graph Convolutional Module (STAGCN) to explicitly model the dynamic spatial dependencies between metro stations, and a Long Short-Term Memory Network (LSTM) to capture intricate local temporal dynamics (periodicity, trends) around each station. This hybrid architecture represents a key advancement by demonstrating how effectively combining station-level temporal modeling with network-level spatial relational learning can significantly improve the accuracy, especially for long-term forecasting, of a particularly challenging and operationally relevant demand type (metro connections), overcoming limitations of both pure GCNs and traditional sequential models in this context.

(3) Empirical Validation and Insight: Through experiments on Shenzhen's shared bike and metro datasets, we demonstrate that our model achieves a coefficient of determination ($R^2$) of 0.889, outperforming baseline models by 6.7% in prediction accuracy. This provides a new theoretical framework for optimizing shared bike resources and planning sustainable urban transportation.

The remainder of this paper is structured as follows: Section 2 presents the literature review. Section 3 describes the research methodology. Section 4 details the experimental design and results analysis. Section 5 concludes the paper.

## 2 Related work

### 2.1 Research on bike-sharing metro connectivity trip identification

The primary task in predicting bike-sharing demand for metro connectivity is to accurately identify such trips. As no platform synchronously records residents' combined bike-sharing and metro trips, obtaining these combined trip records is impossible [10]. Thus, extracting such data is the first step.

This study divides bike-sharing metro connectivity trips into two types. As shown in Fig 1, Type one is inbound trips ending at metro stations, where users bike to a station's buffer zone, park, and transfer to the metro. Type two is outbound trips starting from metro stations, where users transfer to bike-sharing after exiting the metro and ride to their destinations from the buffer zone.

Current studies often use metro stations as the center and set regular buffer zones of 100-500m to identify bike-sharing metro connectivity trips [11–14]. For example, Fan et.al. [5] and Hu et al [15] considered trips with bike-sharing origins or destinations within 300m of a metro station as connectivity trips. In the research of OFO bike-sharing company, orders

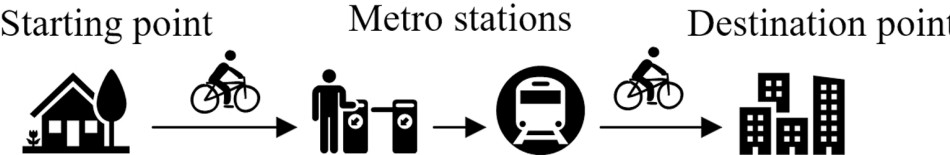

**Fig 1. Schematic diagram of the shared bike connecting to the metro.**

starting or ending within 100 m of a metro station exit are defined as metro – connection trips. Li et al. [16] set a 100m circular buffer zone around metro stations as the identification area. Wu et al.et al. [17] propose an enhanced two-step floating catchment area (E2SFCA) method integrated with Bayes' theorem to assess metro station-oriented shared bike travel demand, incorporating multiple parameters including temporal constraints, spatial proximity, environmental competition ratios, and Point of Interest (POI) service capacity indices. Although buffer zone analysis is easy to operate, it has limitations. The single spatial criterion may lead to duplicate or invalid trip records, reducing identification accuracy and affecting the analysis of travel characteristics.

Unlike past studies using single-circle buffer zones, this paper adopts the KDTree (K-Dimensional Tree) algorithm. Through constructing a KDTree, it performs efficient nearest-neighbor searches for metro stations near bike-sharing locations and calculates the distances. Then, it defines the optimal radius of metro stations' connectivity buffer zones and filters out valid bike-sharing orders within these zones.

## 2.2 Research on bike-sharing demand prediction

The prediction of bike-sharing demand around metro stations constitutes a classic time series forecasting problem, involving the analysis and prediction of bike-sharing usage patterns. As critical urban transportation hubs, metro stations exhibit pronounced spatiotemporal characteristics in their surrounding bike-sharing demand. This demand is influenced by multifaceted factors, including temporal variations (hour-of-day, day-of-week), spatial environments, and weather conditions. The early research methods for predicting the demand of shared bikes mainly employed on traditional statistical models, such as linear regression, ridge regression and autoregressive integrated moving average (ARIMA) models. These methods assumed that the data were stable and had a linear relationship, but they were unable to capture the non-linear and dynamic characteristics of bike demand influenced by time factors (such as peak hours) and external factors (such as weather). With advancements in big data and machine learning, researchers increasingly adopted machine learning algorithms to capture complex nonlinear relationships and process heterogeneous data types. For instance, Torres et al. [18] utilized an SVM model to predict bike demand near metro stations with promising results, while Seo et al. [19] implemented random forest algorithms incorporating station activity data. Hybrid LSTM-ARIMA models [20], have gained increasing prominence due to their superior capacity to handle complex, nonlinear time series data. However, these models often fail to take into account the spatial dependencies between stations, which limits their ability to model demand patterns centered around cities.

Deep learning approaches have demonstrated exceptional potential in temporal sequence processing and large-scale data analytics. Shi et al. [21] applied LSTM networks alongside conventional machine learning models to predict hourly bike-sharing demand, identifying key influencing factors including humidity, peak hours, and temperature. To address spatial dependencies between metro stations and bike-sharing nodes, graph neural networks (GNNs) have been proposed. Kim et al. [22] developed a GCN-based framework that integrates spatial station interdependencies with temporal patterns and global variables (e.g., weather, time-of-day), achieving enhanced prediction accuracy. Zhou et al. [23] introduced the Spatio-Temporal Bike Demand Prediction (ST-BDP) model, leveraging multi-source data and STGCNs to construct spatial demand graphs that account for geographical influences. Ma et al. [24] proposed a CNN-LSTM-Attention hybrid model

analyzing spatiotemporal distribution patterns under seasonal and meteorological influences. Although these advancements have achieved significant results, most deep learning models still encounter challenges in terms of computational efficiency when making long sequence predictions, especially on subway stations with diverse temporal and spatial dynamics.

Recent studies have focused on integrating spatiotemporal attention mechanisms and graph-based methods to capture complex dependencies. Zhu et al. [25] developed a federated learning framework (FedCGAT) integrating attention-based spatiotemporal GNNs with contrastive learning for noise-resistant feature extraction. Feng et al. [26] designed a Spatio-Temporal Aggregation Graph Neural Network (STAGNN) with dynamic adjacency matrices to capture evolving station connectivity. Yang et al. [27] created the FF-STGCN dual-network model combining multi-scale spatiotemporal features with usage pattern similarity learning. Xing et al. [28] established a spatiotemporal fusion network for metro OD flow prediction via spatiotemporal linkage graphs. Sun et al. [29] considered spatial heterogeneity and proposed a two-stage NARX (Nonlinear AutoRegressive with Exogenous Inputs) model based on periodic decomposition to accurately predict bike-sharing demand in individual hotspot areas. Behroozi et al. [30] significantly improved the accuracy of bike-sharing demand prediction by integrating trajectory data, weather data, and visitation data with a gated graph convolutional network. Ma et al. [31] developed a model combining Temporal Convolutional Network (TCN) and Bidirectional Gated Recurrent Unit (BiGRU), optimized using the Chernobyl Disaster Optimizer (CDO) for hyperparameter tuning. Chen et al. [32] investigated various spatiotemporal influencing factors related to bike-sharing and proposed a multi-source data-driven Local-Global Dynamic Multi-Graph Convolutional Network (LGDMGCN) model for multi-step station-level bike-sharing demand prediction. Ma et al. [33] proposed a spatio-temporal graph attention long short-term memory (STGA-LSTM) neural network framework, which utilized multiple data sets to predict the short-term demand of the bicycle-sharing system at the station level. This model can extract the spatio-temporal information of the bike-sharing system and predict the short-term demand for bike rental and return. Zi et al. [34] proposed a deep graph convolutional network model with temporal attention (TAGCN), which is used to predict the number of bicycles entering and exiting each station. TAGCN not only can simulate the spatial and temporal dependencies between different stations, but also can reflect the influence of different time granularities (hourly, daily and weekly time periods). It can effectively capture the dynamic temporal correlation and comprehensive spatial patterns of bicycles entering and exiting stations. These models excel in capturing spatial heterogeneity, but they often require a significant amount of computing resources, which limits their practical application. Moreover, few studies have focused on the specific challenges associated with demand forecasting for shared bikes connected to subways, such as distinguishing connection trips and effectively modeling dependency relationships.

### 2.3 Research gaps

Despite significant progress, several gaps persist in the literature:

(1) Most studies rely on simple buffer methods, failing to filter out disconnected trips, which affects the purity of the data and the accuracy of demand forecasting.

(2) The existing models have deficiencies in capturing complex spatial relationships and multi-scale temporal features.

(3) Models specifically optimized for the spatio-temporal heterogeneity of subway connection scenarios are relatively lacking.

The research on bike-sharing demand prediction has evolved continuously, with traditional statistical methods gradually being replaced by machine learning and deep learning approaches. Deep learning methodologies that integrate spatio-temporal characteristics and multimodal data fusion have demonstrated significant improvements in prediction accuracy and robustness.

## 3 Methods

### 3.1 Connectivity trip identification

Building upon existing methodologies for connecting trip identification, this study implements a KDTree spatial search algorithm. The algorithm constructs a KDTree data structure that maps metro station coordinates into a binary tree-based spatial partitioning system, achieving efficient nearest neighbor searches with optimized time complexity $O(\log_2 n)$. This approach enables rapid identification of the closest metro stations to shared bike GPS origin/destination points [35]. By establishing K-dimensional tree spatial indices for K-dimensional data points, the algorithm performs accelerated proximity matching: input coordinates are efficiently traversed through the tree structure to locate nearest-neighbor metro stations, subsequently calculating the Euclidean distances between trip origin/destination points and their matched stations.

Import bike-sharing and metro station data into ArcGIS. Create multi-ring buffers around each metro station at 50m intervals from 50m to 500m. Use intersect analysis to find bike-share orders within these buffers, and count order changes (including station-entry and station-exit orders) at each radius.

The optimal buffer radius for metro station connecting is determined based on the inflection point of order quantity growth patterns [36]. When the growth rate of order volume begins to decelerate, the corresponding distance threshold is identified as the optimal connecting radius. Concentric buffer zones are established around metro stations, systematically filtering bike-sharing trips whose origin/destination coordinates fall within the optimal buffer range. To authenticate connecting trips, the methodology verifies spatial correspondence between trip endpoints and their nearest metro stations. If the nearest stations to both the origin and destination are identical, the trip is classified as a connecting trip. For non-matching pairs, distance differentials between actual trip endpoints and their nearest stations are calculated. Trips with differentials below 200 meters (equivalent to a 1-minute cycling distance) qualify as connecting trips. Larger differentials trigger metro line collinearity checks: trips connecting non-collinear stations within 2 kilometers (encompassing 90% of verified connecting distances) are classified as connecting trips, while those linking collinear stations are excluded as metro substitution behaviors.

This multi-stage filtering mechanism effectively reduces interference from non-connecting trip records. Experimental results demonstrate significant improvements in connecting trip identification accuracy, validating the methodological efficacy through enhanced precision in distinguishing genuine metro-bike intermodal trips from alternative travel patterns.

### 3.2 Demand forecast for shared bikes facing connection

**3.2.1 Informer model.** The Informer model represents an innovative time series forecasting approach specifically designed to address challenges in long-sequence prediction tasks. By synergistically integrating the advantages of Transformer architectures with the processing capabilities of traditional Recurrent Neural Networks (RNNs), Informer effectively reduces Graphics Processing Unit (GPU) resource consumption while enhancing the practicality and efficiency of extended sequence forecasting. The architectural configuration of the model is illustrated in Fig 2.

The unique advantages of Informer model are as follows:

(1) Multi-head prosparse self-attention mechanism: The Informer model consists of an encoder and a decoder. In the encoder, a multi-head prosparse self-attention mechanism is put forward as a replacement for the traditional multi-head self-attention mechanism. It can train variable feature information in different spaces in parallel. The formula for prosparse attention is as follows:

$$A(Q, K, V) = Soft\max\left(\frac{\bar{Q}K^T}{\sqrt{d}}\right)V$$

(1)

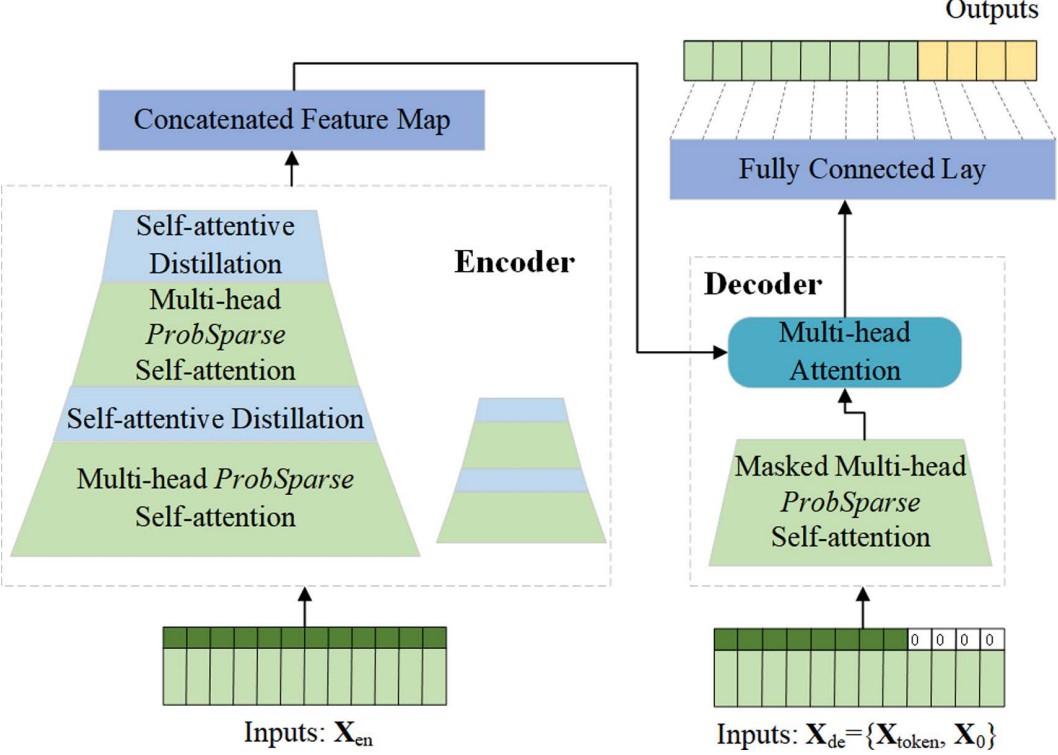

**Fig 2. Informer model structure diagram.**

Where $Q$ is the Query vector, $K$ is the Key vector, $V$ is the Value vector, $d$ is the input dimension, and Softmax is the activation function. The prosparse attention mechanism can effectively reduce the spatial complexity of $O\left(L^2\right)$, without significantly losing accuracy.

(2) Self-attentive distillation: The encoder of the Informer model aims to capture dependencies in long-sequence inputs. But during encoding, feature mapping may produce redundant information. To reduce the high spatial complexity and memory consumption caused by long inputs, the encoder optimizes feature selection through a distillation mechanism, focusing more on key high-level features. It assigns higher weights to important features and reduces the input length layer by layer, lessening the computational and storage burden. The length of the sequence is halved from layer $j$ to $j+1$.

$$X_{j+1}^t = MaxPool\left(ELU\left(Convld\left(\left[X_j^t\right]_{AB}\right)\right)\right)$$

(2)

$X_{j+1}^t$ denotes the distillation operation from layer $j$ to $j+1$, $\left[X_j^t\right]_{AB}$ represents basic variables of the attention module and prosparse attention, MaxPool is the maximum pooling with a stride of 2, ELU is the activation function, and Conv1d is the 1D convolution. After processing by the multi-head prosparse attention mechanism, the sequence data is directly fed into the feature extraction layer, reducing network parameters and highlighting important features. This process involves dimensionality reduction and memory-consumption reduction, compressing the output dimension to half the original length and time range. This operation is repeated until the final output is obtained, which is then passed to the subsequent multi-head attention module for feature interaction. The structure of the self-attentive distillation mechanism is shown in Fig 3.

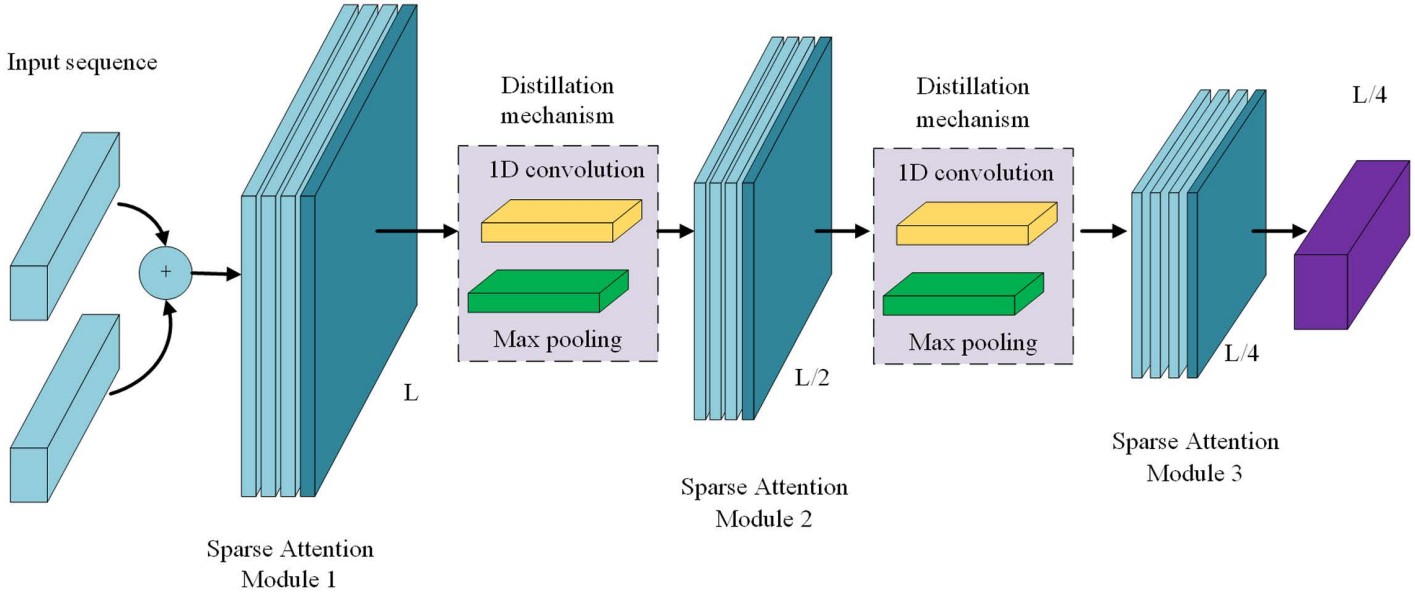

**Fig 3. Self-attentive distillation mechanism.**

(3) Generative Decoder: The decoder is designed to avoid the interference of autoregressive training while maintaining computational efficiency, especially in long-sequence prediction tasks.

The input sequence $X_{de}^t$ of the decoder consists of two parts: the historical sequence $X_{token}^t$ from the encoder's output, which contains historical information of the time series, and the target placeholder sequence $X_0^t$, which is usually zero-padded to prevent information leakage in the autoregressive process and acts as a placeholder for future predictions to ensure that information not yet generated is not used. The formula is as follows:

$$X_{de}^t = Fusion\left[Concat\left(X_{token}^t, X_0^t\right)\right] \in R^{(L_{token}+L_y)d_{model}}$$

(3)

The concatenation operation combines the historical sequence from the encoder and the target placeholder sequence to form an input sequence containing historical and future placeholders. Fusion is a feature fusion operation for further processing of the concatenated information. In the Informer model, the generative decoder uses the results processed by multi-head attention, passes them to a fully connected layer to adjust the dimension, and performs inverse normalization. It adopts a generative reasoning method, generating the entire forecast sequence in one go during training without relying on previous outputs, which greatly improves the computational efficiency of the decoding process. Specifically, the decoder generates complete predictions in one go by processing the entire target placeholder sequence using multi-head attention and fully connected layers, without iteratively depending on historical predictions.

**3.2.2 STAGCN module.** Spatio-Temporal Attention Graph Convolutional Network is a spatio-temporal modeling method combining Graph Convolutional Network and Attention Mechanism. It's tailored for complex spatio-temporal graph data, especially tasks with dependencies in both space and time. The key feature of STAGCN modules is using graph convolutions to capture spatial dependencies, and adding adaptive attention to enhance the interaction modeling of spatial and temporal features. Each metro station is a graph node, and the traffic network between stations (like metro lines and road connections) forms spatial dependencies. STAGCN captures these dependencies through graph convolution and

attention mechanisms, providing valuable spatio-temporal features for subsequent bike-sharing demand prediction.The structural diagram is shown in .

(1) Graph convolution

In bike-sharing demand prediction at metro stations, STAGCN uses graph convolution to capture spatial dependencies. Each station is a node in a graph, with connections like distances forming edges. Each station can be regarded as a node in the graph. The edges between nodes are determined based on two aspects: if two stations are directly connected on the subway line, the corresponding element in the adjacency matrix takes the value of 1; if the spatial distance between the two stations (calculated as the Euclidean distance based on latitude and longitude) is less than the preset threshold of 500 meters, then the corresponding element in the adjacency matrix is also set to 1. For pairs of nodes that meet the above conditions, the elements in the adjacency matrix are 1, indicating spatial dependence; otherwise, they are 0. For a graph $G = (V, E, A)$ with node set V and edge set E, and $A \in \mathbb{R}^{N \times N}$ represents the adjacency matrix, the graph convolution operation is defined as $H^{(l+1)} = \sigma \left( \hat{A} H^{(l)} W^{(l)} \right)$, where $H^{(l)}$ is the node feature matrix at layer $l$, $H^{(0)}$ is the input features, $\hat{A}$ is the normalized adjacency matrix. To eliminate the influence of node degree differences, the symmetric normalization method is employed to process the adjacency matrix: $\hat{A} = D^{-\frac{1}{2}} A D^{-\frac{1}{2}}$, with D the degree matrix, $W^{(l)}$ is the weight matrix at layer $l$, and $\sigma$ is the ReLU activation. GCN aggregates nodes in the spatial dimension via convolution operations based on these connections, capturing the influence and interdependencies between stations.

(2) Spatiotemporal attention mechanism

The STAGCN module handles complex spatio-temporal interactions by modeling spatial and temporal dependencies simultaneously. It can adjust weights adaptively according to the features of different nodes and times, enabling the model to focus on more important spatio-temporal features.

Spatial attention: A metro station's demand is influenced by its surrounding area. STAGCN assigns each station an attention weight via a spatial attention mechanism, automatically learning which spatial features are more important for bike-sharing demand prediction. For example, some stations are linked to commercial areas, while others are near residential zones. The spatial attention mechanism helps the model focus on these diverse environmental features.

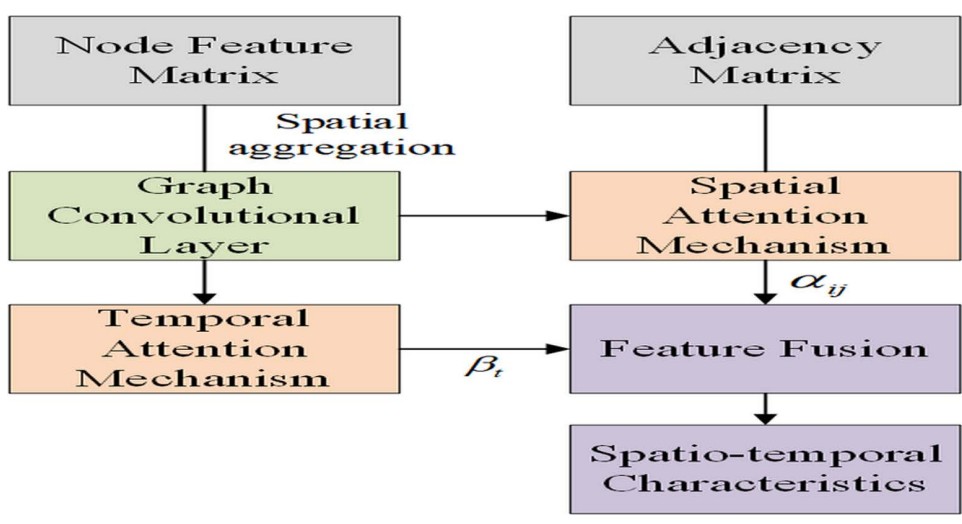

**Fig 4. Structural schematic diagram.**

Assuming we have a node feature matrix $H \in \mathbb{R}^{N \times d}$, where $N$ is the number of nodes and $d$ is the feature dimension, the spatial attention mechanism calculates attention weights $\alpha_{ij}$ for each node pair $(i, j)$ using:

$$\alpha_{ij} = \frac{e^{\left(LeakyReLU\left(a\left[Wh_i \| Wh_j\right]\right)\right)}}{\sum_{k \in N_i} e^{\left(LeakyReLU\left(a\left[Wh_i \| Wh_j\right]\right)\right)}} \tag{4}$$

where $W$ is the linear transformation matrix for feature mapping, $h_i$ and $h_j$ are the feature vectors of nodes $i$ and $j$, $\|$ denotes concatenation, $N_i$ is the set of neighbors of node $i$, and $a$ is the learned attention vector.

Based on the calculated spatial attention weights $\alpha_{ij}$, we perform a weighted aggregation of the neighboring nodes: $h'_i = \sum_{j \in N_i} \alpha_{ij} Wh_j$. This way, the node $i$ new feature pays more attention to its more important neighboring nodes in space.

Temporal Attention: Bike-demand changes over time, especially influenced by different periods near metro stations (e.g., rush hours, weekdays, weekends). Temporal attention enables the model to adjust feature weights according to time steps, adapting to temporal changes and demand patterns. For each time step $t$, the attention weight $\beta_t$ is calculated by:

$$\beta_t = \frac{e^{\left(LeakyReLU(b[Uh_t])\right)}}{\sum_{t'=1}^{T} e^{\left(LeakyReLU(b[Uh_{t'}])\right)}} \tag{5}$$

where $U \in \mathbb{R}^{d' \times d}$ is the linear transformation matrix for time features, it projects the input feature $h_t$ from the dimension $d$ to a new feature space dimension $d$' (where $d$' is the latent dimension selected by the attention mechanism). $h_t \in \mathbb{R}^d$ is the input feature vector at time step $t$ (including the historical demand at that time, time characteristics, and weather characteristics), and $b \in \mathbb{R}^{d'}$ is the learned attention vector, it acts on the transformed feature $Uh_t$ and calculates a scalar "energy" value (through the LeakyReLU activation). $U$ and $b$ are shared among all time steps $t$ and all spatial nodes (metro stations).

Spatio-temporal attention mechanisms can predict bike-sharing demand more precisely by learning weighted combinations of spatial and temporal features.

### 3.2.3 Demand prediction model for shared bikes facing connection.

This paper integrates the models of STAGCN, LSTM, and Informer models to boost bike-sharing demand prediction by leveraging their strengths in spatio-temporal feature extraction, long-dependency modeling, and time-series forecasting. Bike-sharing demand at metro stations is influenced by historical data and surrounding stations' demand changes. The STAGCN module models relationships between stations using spatial graph convolutions. Building on the spatial features from STAGCN, Informer and LSTM are used in parallel.

The Informer models global time-series by focusing on historical data's most relevant information through self-attention, capturing demand fluctuations over long periods and periodic changes between weekdays and weekends. Its prosparse self-attention and encoder enhance prediction efficiency and model performance.Long Short-Term Memory networks represent a specialized variant of Recurrent Neural Networks. Through the strategic integration of gating mechanisms—including forget gates, input gates, and output gates—LSTM effectively mitigates the vanishing gradient problem [37]. These gated architectures enable selective retention and discarding of sequential information, capturing both short-term dependencies and dynamic temporal variations in sequence data. Furthermore, LSTM facilitates localized temporal modeling of preprocessed spatiotemporal features through its recurrent processing paradigm. Finally, the outputs of Informer and LSTM are concatenated and fused through a fully connected layer. By considering spatial relationships, global temporal features, and local fluctuations, our model achieves high-precision and reliable predictions while maintaining computational efficiency. Fig 5 illustrates the model architecture.

   

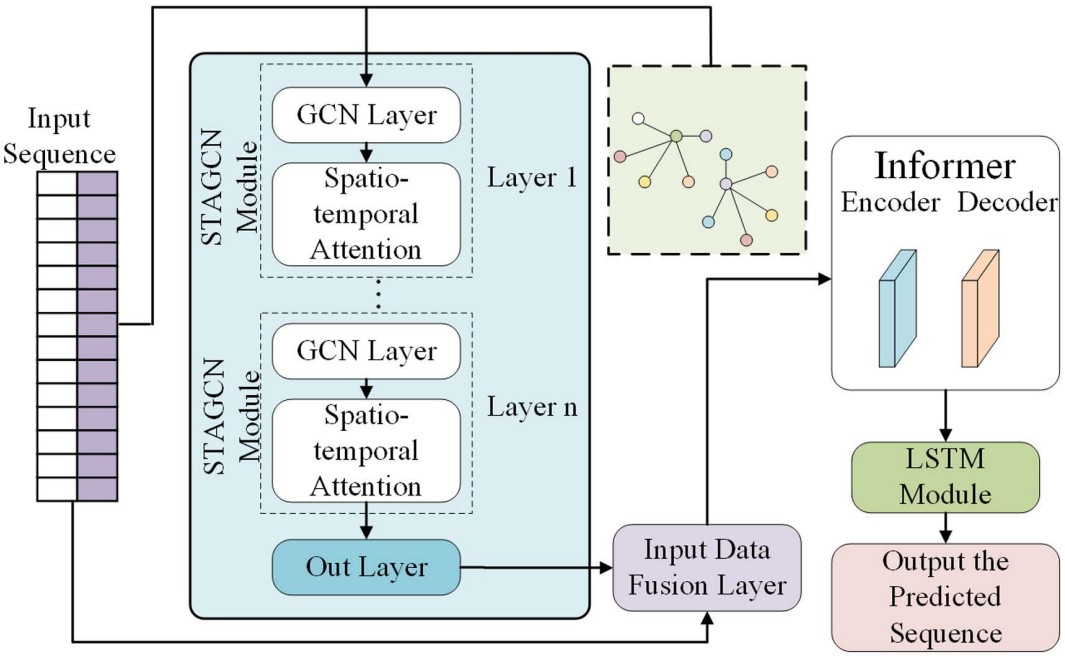

**Fig 5. STAGCN-LSTM-Informer model architecture.**

## 4 Experimental and analysis

### 4.1 Experimental environment and data

The datasets utilized in this study, including shared bike trip records and metro network data, were sourced from the Shenzhen Municipal Government Open Data Platform. Data acquisition was conducted via Python scripts accessing official APIs. The research incorporates 86,907,511 shared bike trip records spanning 61 consecutive days from May 1, 2021 to June 30, 2021. During this period, Shenzhen's metro system maintained 11 operational lines comprising 236 stations. Meteorological data were retrieved from online sources through Python-based web crawlers.

### 4.2 Data preprocessing

Each shared bike order data point includes user ID, trip start time, and origin/destination latitude-longitude coordinates. These raw data require preprocessing steps such as coordinate conversion, data cleaning, and feature extraction before utilization. The preprocessing workflow involves converting all spatial data to a unified coordinate system, removing null or duplicate records based on user ID, filtering out trips outside the study area, and eliminating anomalous data in temporal and spatial dimensions. Specifically, trips with durations shorter than 1 minute or longer than 1 hour, distances less than 100 meters or exceeding 5 kilometers, or those initiated outside Shenzhen metro operational hours (6:00–23:00) are excluded. Ultra-short trips may indicate faulty vehicle scans or GPS anomalies, while excessively long trips likely represent non-connecting behaviors such as recreational cycling. After rigorous filtering, 39,362,644 valid shared bike trip records were retained.

To validate the connecting trip identification method, Fig 6 illustrates the incremental changes in shared bike distribution across buffer zones using a 10-day dataset (May 1–10, 2021) in Shenzhen. The visualization demonstrates the spatial-temporal dynamics of bike accumulation relative to metro station proximity, empirically supporting the buffer radius optimization process.

                                                                                 

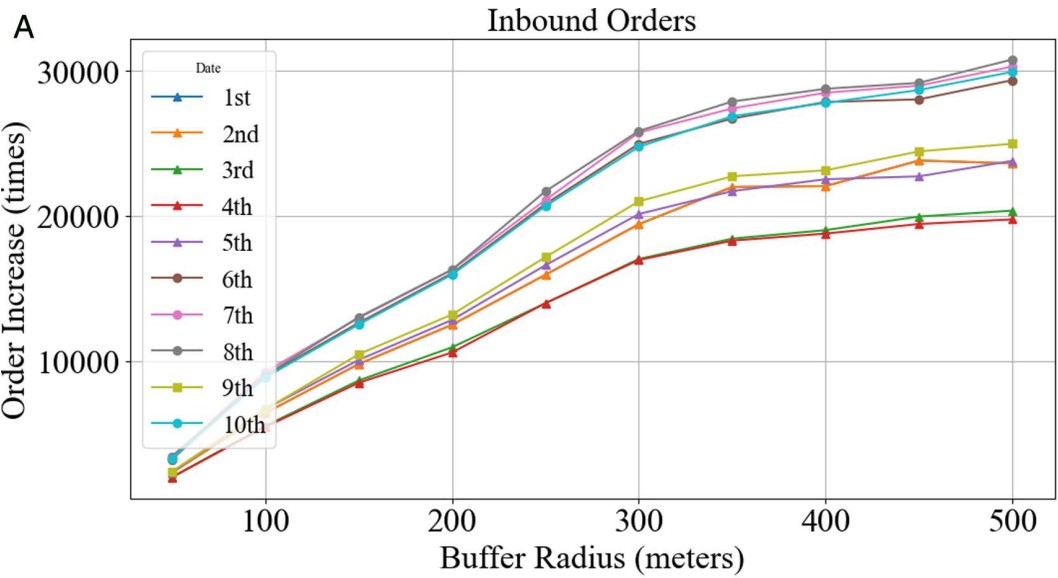

A

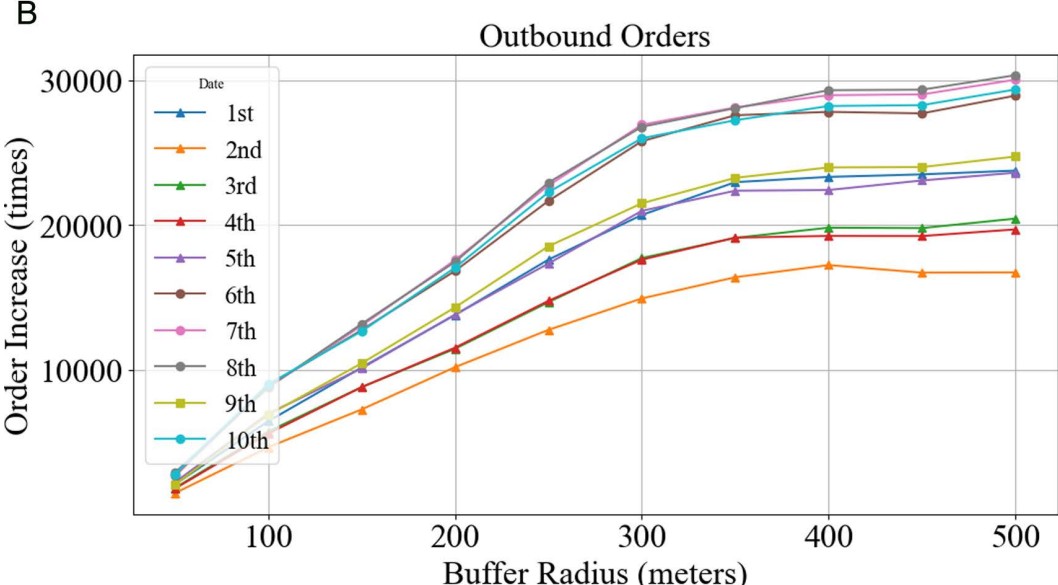

B

**Fig 6. Inbound and outbound orders increased.**

From Fig 6, it can be observed that, whether for inbound or outbound, after 300 meters, the growth rate of orders begins to slow down. Moreover, the rate of change (slope) of the total order growth relative to the buffer radius was calculated using the numerical differentiation method. The results are shown in Fig 7a, where the slope significantly decreases near 300 meters, indicating that the order growth rate begins to stabilize. Additionally, to evaluate the robustness of the 300-meter radius, the order volume within the range of 250 meters to 350 meters was statistically analyzed at intervals of 10 meters, and the percentage change relative to the order volume at 300 meters was calculated. The results in Fig 7b show that the increase in order volume is relatively small after 300 meters. Thus, 300 meters is the optimal connection radius for connecting with subway travel.

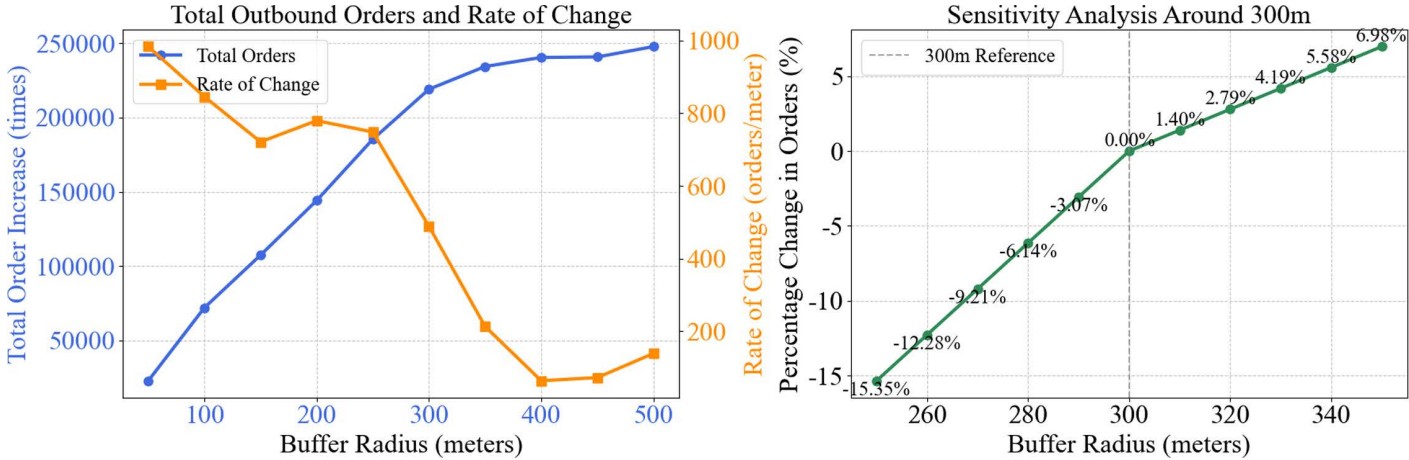

**Fig 7. (a) Order growth volume and growth rate, (b) Sensitivity analysis near 300 meters.**

To enhance the rigor of the identification method in this paper, this study developed a method for constructing the catchment area using Voronoi diagram, and compared the existing research's fixed 300-meter buffer zone method as the benchmark with the method proposed in this paper. Based on the subway stations, Voronoi diagram were constructed to spatially divide the subway stations in Shenzhen. The start and end points of the shared bicycle orders were matched to the corresponding Voronoi diagram regions through spatial connections. If the starting point and the ending point of the shared bicycle belonged to different subway stations (and were not collinear stations), they were marked as transfer orders. Regarding the boundary effect of Voronoi diagram, the orders within a 50-meter range of the intersection area of the polygons were subjected to KDTree nearest neighbor secondary verification to ensure the accuracy of spatial attribution.

Experimental results derived from partial trip data analysis (Table 1) identify 310,816 validated connecting trips, accounting for 22.15% of total bike-sharing orders. Although the Voronoi diagram method avoids the rigid constraint of a fixed radius, its spatial division does not take into account the actual attenuation of connection distances (such as when the polygon of a suburban site is too large), thus the accuracy of the connection orders identified by it is not high. Compared with the 300-meter buffer identification method used in previous studies, the method proposed in this paper filters out 24% of non-connecting travel data, compared with the Voronoi diagram method, this paper has eliminated 26% of non-connecting trips. effectively reducing non-connecting travel within the 300-meter buffer zone, and significantly improving the data purity within the connecting area.

A total of 3,697,465 validated historical connecting trip records were extracted across two consecutive months. For each station's connecting trip data, the Python Pandas package was employed to filter inbound and outbound

**Table 1. Identification result of connecting shared bike to metro.**

| Date | Baseline (300 m buffer) | Voronoi | Proposed | Efficiency ratio |
|------|------|------|------|------|
| 2021/5/6 | 84567 | 87923 | 64855 | 76.69%(Baseline), 73.76%(Voronoi) |
| 2021/5/7 | 85382 | 89456 | 65386 | 76.58%(Baseline), 73.09%(Voronoi) |
| 2021/5/8 | 84587 | 86581 | 65073 | 76.93%(Baseline), 75.16%(Voronoi) |
| 2021/5/9 | 66575 | 68253 | 50504 | 75.86%(Baseline), 73.99%(Voronoi) |
| 2021/5/10 | 86768 | 88776 | 64998 | 74.91%(Baseline), 73.21%(Voronoi) |
| Total | 407879 | 420989 | 310816 | 76.21%(Baseline), 73.82%(Voronoi) |

bike-sharing trips within metro connecting buffer zones during operational hours (06:00–23:00), with trip volumes subsequently aggregated at 15-minute temporal granularity. The dataset incorporates daily meteorological records and temporal features including weekday/weekend/holiday indicators. Using inbound trip volumes within station-specific buffer zones as the predictive target, this study forecasts bike-sharing demand around Shenzhen metro stations. The dataset was partitioned into training, validation, and test sets following a 7:1:2 ratio.

### 4.3 Selection and analysis of influencing factors based on experimental environment

The Pearson correlation coefficient was used to measure the linear relationships between bike-sharing demand (Target) around metro stations from May 1, 2021 to June 30, 2021 and meteorological features (temperature, wind speed, precipitation) as well as temporal characteristics (hourly intervals, weekdays, weekends, holidays). As shown in Fig 8. And the Table 2 presents the collinearity analysis among various features.

From the results, it can be seen that there is a certain correlation between the demand (Target) of shared bikes and various characteristics. Firstly, the correlation between demand and time period is the strongest. The absolute value of the Pearson correlation coefficient is 0.56, indicating that the demand for shared bikes varies significantly at specific times of the day (such as morning and evening rush hours). The demand is usually higher during peak hours, reflecting people's reliance on shared bikes during the rush hours of commuting. Therefore, time period is a key factor and should be given priority attention when predicting the demand for shared bikes.

Secondly, the correlation between demand and temperature is relatively significant. The absolute value of the Pearson correlation coefficient is 0.21, indicating that an increase in temperature may slightly reduce the demand for shared bikes. Especially in hot weather, some users may prefer air-conditioned vehicles or other transportation methods, reducing the

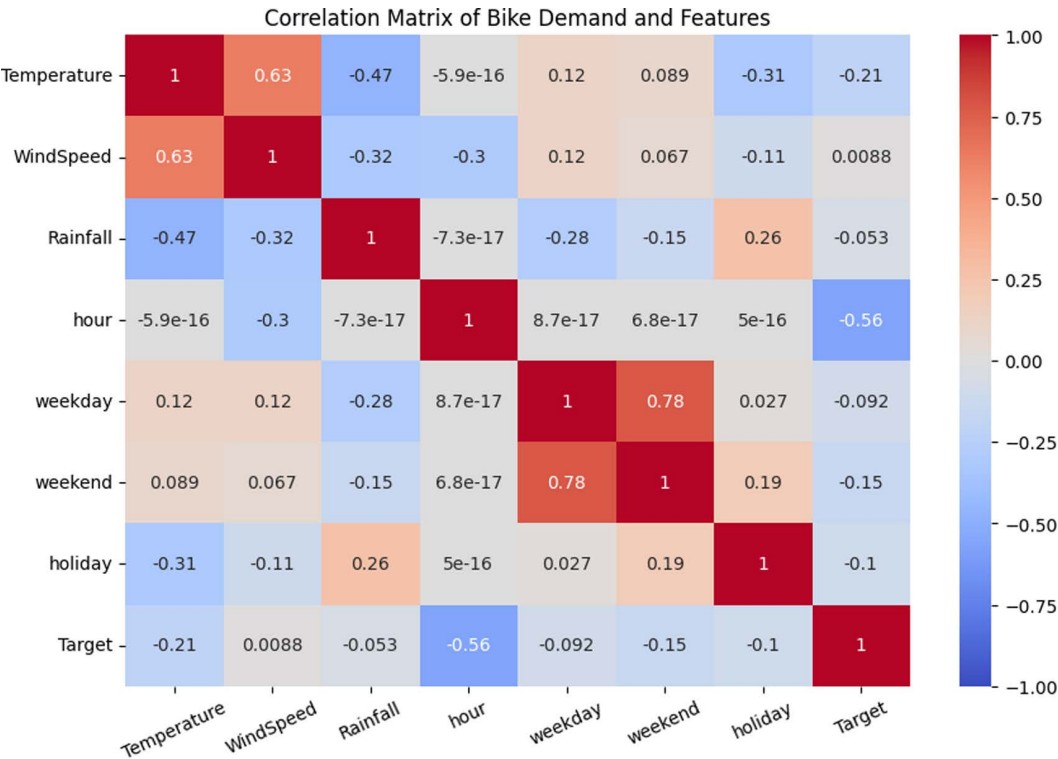

**Fig 8. Correlation of meteorological features.**

**Table 2. Variance Inflation Factor (VIF) analysis for features.**

| Feature | VIF |
|---|---|
| Temperature | 2.203 |
| WindSpeed | 1.864 |
| Rainfall | 1.805 |
| Hour | 4.145 |
| Weekday | 7.074 |
| Weekend | 3.668 |
| Holiday | 1.441 |

demand for bikes. Therefore, the role of temperature in demand prediction cannot be ignored, especially in regions with significant seasonal changes.

The demand for shared bikes is also affected by weekends. The correlation coefficient is 0.15, indicating that the demand is lower on weekends and higher on weekdays. This trend may be related to the different time and purposes of people's outings on weekends. On weekdays, especially during working hours, users have a more concentrated demand for shared bikes, while on weekends, they tend to prefer leisure and short-distance travel, which may lead to a decrease in demand.

Table 2 shows that the VIF of "weekday" is 7.07, indicating the presence of multicollinearity. Therefore, this variable can be excluded. The correlation between wind speed and demand is relatively weak, with an absolute value of the correlation coefficient being 0.008. This indicates that there is a very small correlation between wind speed and the use of shared bikes, and it can be ruled out. As for holidays and rainfall, the absolute values of the Pearson correlation coefficients are 0.1 and 0.053 respectively. This suggests that rainy days and holidays have a certain inhibitory effect on the use of shared bikes, but the impact is relatively small. Therefore, in the subsequent experiments, the characteristics such as the selected time period, temperature and weekend were chosen as the influencing factor characteristics. Fig 9 is a scatter plot of the correlation features.

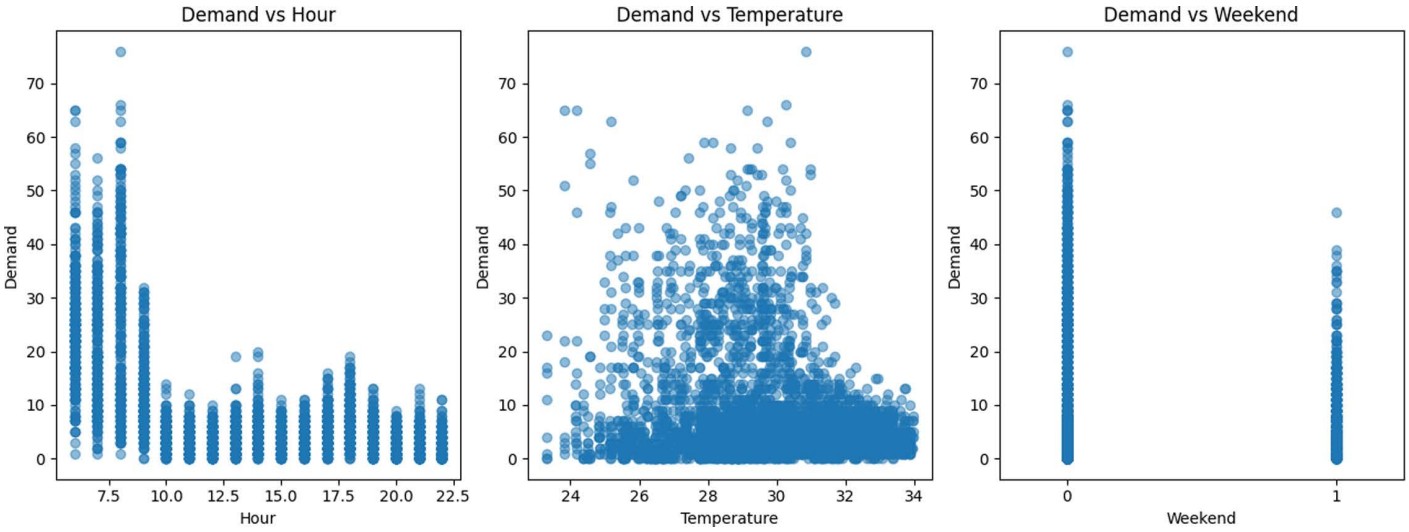

**Fig 9. Correlation feature scatter plot.**

## 4.4 Model parameters and evaluation indicators

The model was trained with a batch size of 64 using the Adam optimizer (learning rate = 0.0001) to facilitate rapid convergence. To mitigate overfitting, an early stopping strategy (patience = 20 epochs) was implemented, terminating training when validation loss plateaued. A sliding window approach generated continuous temporal sequences, utilizing historical data from the preceding 3 days (window size = 204 timesteps) to forecast 1-day ahead demand (prediction length = 68 timesteps).

The regression model used in this paper to evaluate the index is: Mean Absolute Error (MAE), Root Mean Squared Error (RMSE) and R-Square ($R^2$), To evaluate the performance of the prediction model in the task of predicting the demand for shared bikes around the metro.

$$MAE = \frac{1}{N} \sum_{i=1}^{N} |\hat{y}_i - y_i|$$

(6)

$$RMSE = \sqrt{\frac{1}{N} \sum_{i=1}^{N} (\hat{y}_i - y_i)^2}$$

(7)

$$R^2 = 1 - \frac{SS_{res}}{SS_{tot}}$$

(8)

Where $y_i$ denotes observed values, $\hat{y}_i$ represents predicted values, and $N$ indicates the sample size of the test set.

## 4.5 Comparison and ablation test results analysis

(1) Comparative experiment

To validate the efficacy of the proposed model, comparative analyses were conducted against baseline approaches including Gradient Boosting (XGBoost), LSTM, STAGCN [38], Transformer architectures, and hybrid models (CNN-LSTM, SARIMA-LSTM). As demonstrated in Table 3, the proposed model consistently outperforms conventional single models like XGBoost. Compared to the CNN-LSTM benchmark, our framework achieves reductions of 0.77 in RMSE and 0.699 in MAE. The model attains an R² score of 0.893, with RMSE and MAE values reaching 4.021 and 2.531 respectively, substantiating its superior predictive accuracy among existing time series forecasting methodologies.

(2) Ablation experiment

Ablation experiments are conducted with the Informer model as the baseline to verify the contribution and effectiveness of each module. And compare the performance of the LSTM and Informer model parallel fusion architecture and sequential stacking architecture in this paper, thereby verifying the superiority of the parallel fusion architecture proposed in this

**Table 3. Comparative experimental performance evaluation.**

| model | RMSE | MAE | R² |
|---|---|---|---|
| XGBoost | 6.256 | 3.483 | 0.710 |
| LSTM | 6.161 | 3.437 | 0.721 |
| Transformer | 5.252 | 3.355 | 0.783 |
| STAGCN | 4.588 | 3.088 | 0.837 |
| SARIMA-LSTM | 4.610 | 3.255 | 0.825 |
| CNN-LSTM | 4.791 | 3.230 | 0.794 |
| STAGCN-LSTM-Informer | **4.021** | **2.531** | **0.893** |

paper. As shown in Table 4, the prediction accuracy improves with the addition of each module, indicating that all modules are essential for accuracy enhancement. Adding the LSTM module reduces the model's MAE by 0.355. Incorporating the STAGCN module lowers the RMSE and MSE by 0.353 and 0.449 respectively, showing it boosts prediction accuracy. This model outperforms the sequential stacking architecture and other baseline models in terms of RMSE (4.036), MSE (2.692), and R2 (0.889). The output of STAGCN is processed in parallel by LSTM and Informer models, and the features of different models are fused through a fully connected layer, which can better capture patterns at different time scales, improve the accuracy and robustness of the prediction, while the sequential stacking architecture may suffer from performance degradation due to information compression. Fitting results of a two-day comparison experiment on the test set are visualized. Fig 10 indicates that our model's prediction curve fits better than other models, proving its superior bike-sharing demand prediction capability.

(3) Efficiency comparison

In order to verify the computational efficiency of the Informer model, this study compared the FLOPs (floating point operations) of different models in the same task and the seconds per epoch. The following Table 5 shows the computational efficiency of different models. The FLOPs of the Informer model (310.284M per round) is approximately 35% lower than that of the Transformer model (480.251M per round). Thanks to its ProbSparse Attention mechanism, the complexity has been optimized from $O\left(L^2\right)$ to $O\left(L \log L\right)$, verifying the advantage of the Informer model in computational efficiency. It can be observed that compared with the model STAGCN-LSTM-Informer in this paper, the FLOPs value of

**Table 4. Performance evaluation of ablation experiments.**

| model | RMSE | MAE | R² |
|---|---|---|---|
| Informer | 4.619 | 3.280 | 0.822 |
| LSTM-Informer [39] | 4.532 | 2.925 | 0.850 |
| STAGCN-Informer [40] | 4.266 | 2.831 | 0.866 |
| Sequential(STAGCN-LSTM-Informer) | 4.398 | 2.853 | 0.857 |
| Parallel(STAGCN-LSTM-Informer) | **4.036** | **2.692** | **0.889** |

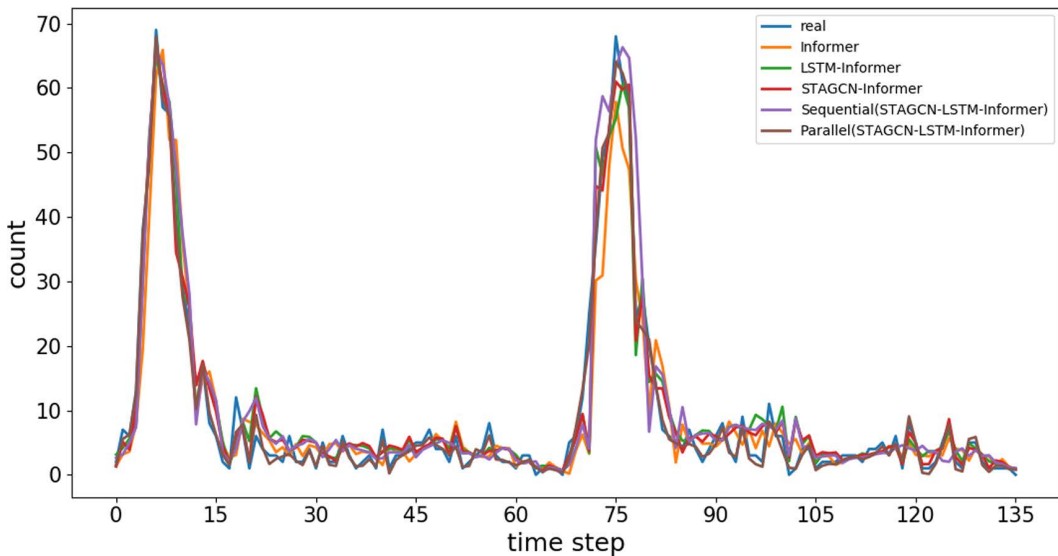

**Fig 10. Prediction results of ablation experiments.**

**Table 5. Comparison of model calculation efficiency.**

| Model | FLOPs(M) | Training Time(s/epoch) |
| --- | --- | --- |
| Informer | 310.284 | 5.23 |
| STAGCN | 532.461 | 9.11 |
| Transformer | 480568 | 6.87 |
| LSTM-Informer | 430.251 | 9.89 |
| STAGCN-LSTM-Informer | 556.195 | 11.41 |

the LSTM-Informer model has decreased by 125.944M, because it does not use the spatio-temporal attention module to capture temporal information and spatial information between stations, thus consuming less energy, but its RMSE has increased by 0.496. Although the model in this paper increases the computational load due to the addition of space-time modeling, it significantly improves the prediction accuracy through parallel processing.

(4) Generalization ability verification

To evaluate the generalization ability of the model, this study added an independent test set (in July 2021, consisting of 1,958,732 shared bike pick-up and drop-off order data) on top of the original dataset, which was used to assess the predictive performance of the model in future time periods. The test results are shown in Fig 11, and compared with the results of the main test set as shown in Table 6. The STGACN-LSTM-Informer model still performed very robustly on the independent test set (R²= 0.872), although it slightly decreased compared to the original training set, it still outperformed the baseline model and maintained a high level of accuracy. This indicates that the model can generalize well to future time periods.

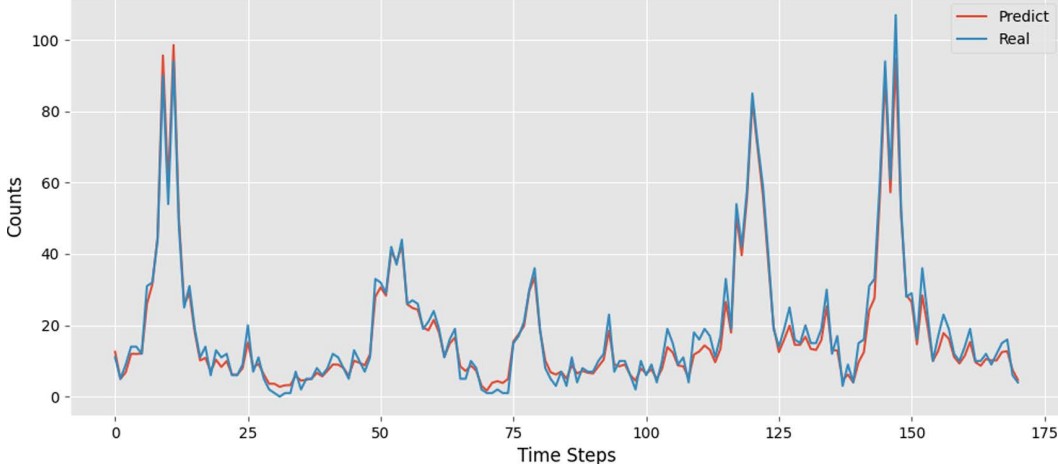

**Fig 11. The prediction results of the independent test set.**

**Table 6. Performance comparison on different test sets.**

| Test Dataset | Model | RMSE | MAE | R² |
| --- | --- | --- | --- | --- |
| Original | STGACN-LSTM-Informer | 4.036 | 2.692 | 0.889 |
| Independent | STGACN-LSTM-Informer | 4.215 | 2.813 | 0.872 |

## 5 Discussion

This study proposes a bike-sharing demand prediction model integrating Spatio-Temporal Attention Graph Convolutional Networks, LSTM, and the Informer architecture, designed to accurately forecast bike-sharing demand dynamics around metro stations. The STAGCN module captures spatial correlations in bike-sharing demand data through graph-based learning, enabling precise identification of demand fluctuations across metro stations. By leveraging parallel processing of Informer and LSTM architectures, the model extracts and processes temporal features across multiple scales. Experimental results demonstrate that this integrated framework achieves superior predictive accuracy and stability compared to conventional models and other time series forecasting approaches in bike-sharing demand prediction tasks.

By accurately predicting the short-distance travel demands in the vicinity of subway stations, bike-sharing companies can optimize their vehicle dispatch strategies, reducing the shortage or idleness of vehicles during peak hours. This not only improves the efficiency of users finding bikes near subway stations but also enhances the overall user experience and urban order management. At the same time, demand prediction technology can be combined with smart city systems. For instance, by integrating real-time passenger flow data with electronic fence technology, the vehicle placement locations can be dynamically adjusted. This not only reduces the operating costs of the companies but also improves resource utilization, promoting the transformation of the bike-sharing industry towards a more refined and intelligent operation model.

Research still faces some limitations and room for future improvement. First, broader validation across diverse urban contexts is required to verify the model's generalization capabilities. Second, advancements in computational power and data availability could enable the integration of more sophisticated network architectures and advanced feature engineering techniques to enhance prediction precision. Finally, future work should investigate real-time data assimilation for dynamic model adaptation and instantaneous demand forecasting.

## Supporting information

**S1 File. Paper code: Pure model code.**
(ZIP)

## Author contributions

**Conceptualization:** Xue Xing, Le Wan.

**Data curation:** Le Wan.

**Formal analysis:** Le Wan.

**Funding acquisition:** Xue Xing.

**Investigation:** Le Wan.

**Methodology:** Xue Xing, Le Wan.

**Project administration:** Xue Xing.

**Resources:** Xue Xing.

**Supervision:** Xue Xing.

**Validation:** Le Wan, Fahui Luo.

**Writing – original draft:** Xue Xing, Le Wan.

**Writing – review & editing:** Xue Xing, Le Wan.

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
