## [Decision Letter · Decision Letter 0]

Dear Dr. Xing,

Thank you for submitting your manuscript to PLOS ONE. After careful consideration, we feel that it has merit but does not fully meet PLOS ONE’s publication criteria as it currently stands. Therefore, we invite you to submit a revised version of the manuscript that addresses the points raised during the review process.

We look forward to receiving your revised manuscript.

Kind regards,

Qing-Chang Lu

Academic Editor

PLOS ONE

Journal Requirements:

3. Thank you for stating the following financial disclosure: [Science and Technology Development Plan Project of Jilin Province (20210101416JC Education Industrial Cultivation Project of Jilin Province (JJKH20230306CY)]. 

4. Thank you for uploading your study's underlying data set. Unfortunately, the repository you have noted in your Data Availability statement does not qualify as an acceptable data repository according to PLOS's standards.

Additional Editor Comments:

I would like to invite the authors to submit a revised version of their manuscript. It is the authors' choice whether to include the references suggested by the reviewers or not.

Reviewers' comments:

Reviewer's Responses to Questions

**Comments to the Author**

1. Is the manuscript technically sound, and do the data support the conclusions?

Reviewer #1: Yes

Reviewer #2: Yes

Reviewer #3: Yes

2. Has the statistical analysis been performed appropriately and rigorously?

Reviewer #1: Yes

Reviewer #2: Yes

Reviewer #3: No

3. Have the authors made all data underlying the findings in their manuscript fully available?

Reviewer #1: Yes

Reviewer #2: No

Reviewer #3: Yes

4. Is the manuscript presented in an intelligible fashion and written in standard English?

Reviewer #1: Yes

Reviewer #2: No

Reviewer #3: Yes

Reviewer #1: The manuscript is technically clear, sound and well written. The data is available online without any restriction. I would like to recommend you to add the graphs of your comparative results along with the tables.

Reviewer #2: The manuscript addresses an important problem by focusing on demand forecasting for shared bicycles around metro stations to improve metro connectivity. The identification of metro-connected trips and the integration of STAGCN, LSTM, and Informer models are valuable technical elements.

However, several areas require improvement:

Literature Review Structure and Analytical Depth:

The literature review mainly describes what existing studies have done but lacks an analytical perspective on research gaps. I recommend restructuring it with a literature review table summarizing prior work’s methods, assumptions, and focus areas, and clearly identifying the gaps. This would better position the specific contribution of integrating metro connectivity within the broader context.

Clarifying the Contribution:

Currently, the contribution is stated more as a description of the work completed rather than highlighting the true research contribution. The paper should explicitly articulate what gap it addresses and how it advances knowledge compared to existing studies on shared bike demand forecasting.

Managerial Insights and Practical Framework:

Providing managerial insights on how operators can use the proposed model, along with a conceptual framework for practical deployment, would strengthen the real-world relevance.

Visualization and Result Analysis:

Result presentation should be improved, both visually and analytically. Beyond graphical comparisons, analyzing key factors influencing prediction accuracy would enhance understanding.

Relevant Studies:

The following papers may help enrich the analysis:

Ma, X., Yin, Y., Jin, Y., He, M., & Zhu, M. (2022). Short-term prediction of bike-sharing demand using multi-source data: a spatial-temporal graph attentional LSTM approach. Applied Sciences, 12(3), 1161.

Yu, L., Feng, T., Li, T., & Cheng, L. (2023). Demand prediction and optimal allocation of shared bikes around urban rail transit stations. Urban Rail Transit, 9(1), 57-71.

Hu, J., Wagner, F., Ayargarnchanakul, E., Nachtigall, F., Milojevic-Dupont, N., Lu, C., & Creutzig, F. Examining Urban Form Effects on Integrated Use Demand of Bike-Sharing and Metro. Available at SSRN 4985690.

Zi, W., Xiong, W., Chen, H., & Chen, L. (2021). TAGCN: Station-level demand prediction for bike-sharing system via a temporal attention graph convolution network. Information Sciences, 561, 274-285.

In summary, while the proposed method is promising, revisions to better structure the literature review, sharpen the articulation of the contribution, and deepen the result analysis are necessary to strengthen the manuscript.

Reviewer #3: This manuscript addresses the important challenge of accurately forecasting dockless bike-sharing demand around metro stations. The proposed methodology is structured as a three-stage pipeline: (1) a KDTree-based approach to identify actual metro-bicycle transfers and determine an optimal buffer radius; (2) a deep hybrid model incorporating Spatio-Temporal Attention Graph Convolutional Networks (STAGCN), LSTM, and Informer components; and (3) empirical evaluation using a two-month dataset from Shenzhen. While the integration of graph learning, attention mechanisms, and long-sequence forecasting is promising, several aspects require clarification, justification, and methodological enhancement.

Major comments:

1- While KDTree is applied to improve over fixed-radius methods, the final model still employs a uniform 300-meter buffer. The added value of KDTree is not fully demonstrated. Why not explore learning flexible, irregular catchment zones from the data? At minimum, comparing the optimal radius found to alternatives like polygonal catchment boundaries could enhance the rigor.

2- The identification of the inflection point on the order-growth curve lacks a formal definition. The method appears heuristic. A sensitivity analysis showing how prediction accuracy varies with small radius changes would strengthen confidence in the selected threshold.

3- The paper does not clearly describe how the spatial adjacency matrix is computed. Is it based on physical metro-line connections, Euclidean distance, demand correlation, or another metric? Explicitly detailing this step is critical for reproducibility and understanding the spatial relationships modeled.

4- While time-of-day, temperature, and weekend indicators are selected based on Pearson correlation (|ρ|>0.15), the exclusion of other weather variables like rainfall and humidity is not justified. Were these omitted due to multicollinearity, missing data, or weak correlations?

5- Eq. 5: The attention mechanism introduces parameters U and b, but their dimensions and roles are not clearly explained. Clarifying whether these are shared across time steps or stations, and indicating tensor shapes, would improve the mathematical transparency.

6- The STAGCN output is processed in parallel by LSTM and Informer models and then merged via an MLP. It is unclear why this architecture is favored over sequential stacking (e.g., LSTM → Informer). An ablation study exploring different fusion strategies could better justify the architectural design.

7- There is a discrepancy between the abstract (RMSE = 4.036, MAE = 2.692) and Table 2 (RMSE = 4.021, MAE = 2.531). Please reconcile these numbers and ensure consistency throughout the manuscript.

8- Although the Informer component is touted for its computational efficiency, no empirical evidence is provided. Including a table comparing training and inference times (e.g., FLOPs, seconds per epoch, GPU hours) across models would substantiate these claims.

9- The use of only 61 days of data for evaluation raises concerns about model generalization. Testing the model on a temporally disjoint holdout set (e.g., a later month) would better assess seasonality effects and the robustness of the predictions.

Minor comments:

a) In table 1 headers. Instead of “300-meter buffer zone identification method” vs. “This paper identifies the method,” maybe go with “Baseline (300 m buffer)” and “Proposed.”

b) Table 2 title typo: “ST A GCN” —> “STAGCN.”

c) Section 4.1: remove the word “programmatically” from “Data acquisition was conducted programmatically via Python scripts…” (it’s redundant).

d) Figure 7’s y-axis needs units.

e) In the Related Work, Informer shouldn’t be in with “traditional” time-series methods, it is a recent Transformer variant.

**Do you want your identity to be public for this peer review?** For information about this choice, including consent withdrawal, please see our Privacy Policy

Reviewer #1: **Yes: ** Muhammad Ahmad

Reviewer #2: No

Reviewer #3: No

---

## [Author Response · Author response to Decision Letter 1]

20 Jun 2025

All the comments from the reviewers and editors have been responded to in the "Response to Reviewers.pdf" document.

---

## [Decision Letter · Decision Letter 1]

Demand Prediction for Shared Bicycles around Metro Stations Incorporating STAGCN

PONE-D-25-16186R1

Dear Dr. Xing,

We’re pleased to inform you that your manuscript has been judged scientifically suitable for publication and will be formally accepted for publication once it meets all outstanding technical requirements.

Kind regards,

Qing-Chang Lu

Academic Editor

PLOS ONE

Additional Editor Comments (optional):

Reviewers' comments:

Reviewer's Responses to Questions

**Comments to the Author**

Reviewer #2: All comments have been addressed

Reviewer #3: All comments have been addressed

2. Is the manuscript technically sound, and do the data support the conclusions?

Reviewer #2: Yes

Reviewer #3: Yes

3. Has the statistical analysis been performed appropriately and rigorously?

Reviewer #2: Yes

Reviewer #3: Yes

4. Have the authors made all data underlying the findings in their manuscript fully available?

Reviewer #2: (No Response)

Reviewer #3: No

5. Is the manuscript presented in an intelligible fashion and written in standard English?

Reviewer #2: Yes

Reviewer #3: Yes

Reviewer #2: The authors have successfully addressed most of my comments, and I find the current version of the manuscript suitable for publication.

Reviewer #3: The authors addressed all my comments. Thank you for the comprehensive responses. My final comment is to just include some recent works that addressed similar problems, e.g., Belkessa, L., et al., 2024, October. Multi-Channel Spatio-Temporal Graph Convolutional Networks for Accurate Micromobility Demand Prediction Integrating Public Transport Data. In Proceedings of the 2nd ACM SIGSPATIAL Workshop on Sustainable Urban Mobility (pp. 5-13).

**Do you want your identity to be public for this peer review?** For information about this choice, including consent withdrawal, please see our Privacy Policy

Reviewer #2: No

Reviewer #3: **Yes: ** Mostafa Ameli

---

## [Editor Report · Acceptance letter]

PONE-D-25-16186R1

PLOS ONE

Dear Dr. Xing,

I'm pleased to inform you that your manuscript has been deemed suitable for publication in PLOS ONE. Congratulations! Your manuscript is now being handed over to our production team.

Kind regards,

on behalf of

Dr. Qing-Chang Lu

Academic Editor

PLOS ONE